# Biochar—A Panacea for Agriculture or Just Carbon?

**Elvir Tenic †, Rishikesh Ghogare † and Amit Dhingra \*** 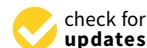

Department of Horticulture, Washington State University, Pullman, WA 99164, USA; elvir.tenic@wsu.edu (E.T.);
rishikesh.ghogare@wsu.edu (R.G.)
\* Correspondence: adhingra@wsu.edu
† These authors contributed equally to this work.

**Abstract:** The sustainable production of food faces formidable challenges. Foremost is the availability of arable soils, which have been ravaged by the overuse of fertilizers and detrimental soil management techniques. The maintenance of soil quality and reclamation of marginal soils are urgent priorities. The use of biochar, a carbon-rich, porous material thought to improve various soil properties, is gaining interest. Biochar (BC) is produced through the thermochemical decomposition of organic matter in a process known as pyrolysis. Importantly, the source of organic material, or 'feedstock', used in this process and different parameters of pyrolysis determine the chemical and physical properties of biochar. The incorporation of BC impacts soil–water relations and soil health, and it has been shown to have an overall positive impact on crop yield; however, pre-existing physical, chemical, and biological soil properties influence the outcome. The effects of long-term field application of BC and how it influences the soil microcosm also need to be understood. This literature review, including a focused meta-analysis, summarizes the key outcomes of BC studies and identifies critical research areas for future investigations. This knowledge will facilitate the predictable enhancement of crop productivity and meaningful carbon sequestration.

**Keywords:** agronomy; sustainability; organic fertilizer; crop productivity; soil acidification; soil organic matter; pyrolysis; microbial activity

---

## 1. Introduction

> *"If you desire peace, cultivate justice, but at the same time cultivate the fields to produce more bread; otherwise there will be no peace"* Norman Borlaug, Oslo, Norway, December 11, 1970. Nobel lecture.

What was prevalent in the 1960s holds true yet again—the world stands at a threshold where the availability of food is threatened, albeit for reasons different than six decades ago. The changing climate, deteriorating land and water conditions, and loss of biodiversity present unprecedented challenges for humankind [1]. At present, greenhouse gas (GHG) emissions are increasing rapidly, with carbon dioxide ($CO_2$) levels rising more than 3% annually since the 2000s. These GHG discharges have a drastic impact on the climate, despite global efforts to reduce the emissions over the last few decades [2]. As a step toward reducing GHG emissions, more than 100 countries signed and ratified the Paris Climate Agreement, aiming to limit the increase in global temperature to 1.5–2 °C over the next 30 years [3]. The achievement of this target requires the swift adoption of carbon-neutral and carbon-negative technologies to limit the global GHG emissions to approximately 9.8 gigatons of carbon [4,5]. Several approaches are being considered for $CO_2$ removal from the atmosphere, such as the adoption of bioenergy, direct carbon capture, afforestation and reforestation, the modification of agricultural practices, the use of bioenergy, and the direct infusion of recalcitrant carbon into the soil using biochar (BC) [6–9]. The longer-term sequestration of carbon into the soil using biochar is one of the potential carbon-negative approaches. As soils store twice as much carbon compared

to atmospheric reserves and for longer periods, it has been hypothesized that increasing global soil organic matter stocks by 4 per 1000 (or 0.4%) per year in agricultural land can offset 30% of global greenhouse emission [4].

The agricultural and industrial revolutions, combined with unsustainable farming practices, have significantly affected global soil health. This is mainly a consequence of the type of fertilizer used in crop production. Earlier practices of using manure and compost replenished the soil organic matter (SOM) on a regular basis. However, the use of petroleum-derived chemical fertilizers is detrimental to SOM as they enhance the accumulation of salt and reduce microbial diversity. Fertilizers derived from the Haber–Bosch process contribute to more than 1% of total global $CO_2$ emissions [10,11]. Soil health has further declined with the gradual acidification of arable lands and continual soil erosion negatively affecting crop yields throughout the world. While the use of compost and manure to enhance and maintain SOM is an option, it presents limitations due to the accumulation of organic pollutants, increased pathogen pressure, and leaching of excess nutrients into waterways, leading to eutrophication [12,13].

There is a long history of enriching soils with recalcitrant carbon practiced by indigenous farmers in different parts of the world. Black-earth-like anthropogenic soils known as 'Terra Preta' have been discovered in several regions of South America and Japan. These dark soils were amended with charcoal-like substances, generally referred to as biochar (BC), and possibly other amendments such as manure, which conferred enhanced fertility to the soil [14,15]. Chemical analysis revealed that the BC-treated areas contained 70 times more carbon than the surrounding soils, demonstrating its long half-life [16]. The enhanced fertility of these soils most likely resulted from increased SOM, higher pH, higher water-holding capacity, and high nutrient-holding capacity [16–18]. Due to the potential advantages of 'Terra Preta', several global efforts are afoot to recreate such soils. Biochar represents an organic soil amendment that improves soil quality for agricultural production [19].

There are various studies reporting the impact of BC on plant growth and development; however, the results have remained inconsistent [20]. There are several types of biochar, produced using various pyrolysis parameters and feedstocks. A standardized biochar with predictable physical, biological, and chemical properties, which is beneficial to plant growth and development, remains to be developed [21]. Furthermore, a positive impact on growth and development does not directly translate to improved yields. In fact, there are reports indicating that enhanced growth and development compromises the plant defense systems. Therefore, this review discusses and summarizes the various aspects of how biochar impacts soil quality and crop productivity. The discussion concludes with a summary of some of the areas that need to be addressed to enable the widespread use of biochar in environmental, economic, agricultural, and ecological contexts.

## 2. Biochar and Soil

### 2.1. Biochar

Biochar (BC) is a carbon (C)-rich, porous material produced during the process of pyrolysis, which involves the thermochemical decomposition of organic matter in an oxygen-limited environment. Any feedstock, such as forest residue, agricultural by-products, and waste biomass can be converted into liquid fuels, gasses, and BC. The properties and yields of BC are highly variable depending on the rate of pyrolysis (fast/slow), feedstock, pyrolysis temperature, and retention time. Generally, slow pyrolysis with a heating rate of 5–20 °C per minute with higher residence time results in higher BC yield [22,23]. Fast pyrolysis with a higher heating rate (>100 °C/min) and lower residence time results in a higher yield of liquid fuel and reduced BC output [24]. Due to the complex nature of pyrolysis and diversity of feedstock, the final chemical and physical properties of BC vary. For example, a recent meta-analysis concluded that BC produced at higher temperature (600–699 °C) had a higher pH of approximately 9 compared to BC produced at lower temperatures (300–399 °C) with an approximate pH of 5 [25]. This observation was supported by another recent meta-analysis [26]. The higher reaction

temperatures reduce the amount of aliphatic carbons, oxygenated functional groups, cation exchange capacity (CEC), and total content of N, H, and O. However, a higher temperature of production resulted in increased pH, amount of C fixed, total ash content, total C, and surface area of BC [25,27,28]. Ultimately, the bulk property and surface characteristics of any BC is determined by the feedstock source along with the pyrolysis parameters [23,29]. There remains a critical need to understand the characteristics of BCs produced from different feedstock, pyrolysis parameters, and the resulting relative impact on soil. In the following sections, recent research on BCs has been collated, and the effects of various BC regimens on soil physical properties, soil–water relations, soil organic matter, microbial activity, soil tilth and nutrient status, pH, crop productivity, biotic stresses, and abiotic stresses have been discussed.

## 2.2. Impact of Biochar on Soil

Physical Properties

Physical properties of soil, such as bulk density, porosity, and water retention are important variables that impact plant growth and development. Human intervention in agricultural practices causes soil compaction, which is one of the key factors affecting plant growth [30]. Soil texture also plays a key role. Soil compaction above 1.7 g cm$^{-3}$ results in restricted root growth and limits access to water and nutrients [31]. As a consequence, the yields of many crops such as soybean and corn have been shown to be negatively impacted [32,33]. The threshold bulk density for impact on root growth varies, with clayey soils having a lower bulk density threshold.

Amending soils with biochar increases soil porosity while decreasing soil bulk density, which aids in the transport of water, nutrients, and gases. These alterations encourage root formation and increased microbial respiration [26].

A meta-analysis reported that the addition of BC to soil reduced the bulk density of the soil by an average of 7.6% and increased its water-holding capacity and porosity by 15.1% and 8.4%, respectively [34]. Similar results were reported in another meta-analysis where the average bulk density was decreased by 12% [35]. Fifteen of the 17 studies conducted in 2019 reported that biochar effectively reduced soil bulk density and increased the porosity and available water content (Table 1). However, there were two studies that reported either no effect on bulk density after the addition of BC [36] or an insignificant decrease [37]. It was also observed that a larger average BC particle size was more effective in reducing the bulk density of sandy loam soil than sandy soil [38]. In the case of sandy soil, bulk density significantly decreased and water-holding capacity was significantly increased with the addition of BC with small particle size [38]. A majority of the recent studies used biochar produced from agricultural residue and woody residue. Generally, a positive effect on the physical properties of soil was reported (Table 1).

**Table 1.** A summary of recent studies related to the impact of biochar on physical properties of soil. The soil types listed in this table correspond to the types reported in the original publication. The usage of term 'slightly' in any of the categories indicates a change in mean values. BC: biochar.

| Exp. Type | BC Feedstock | Pyrolysis Temp. (°C) | Soil Type | Bulk Density | Available Water Content | Total Porosity | BC Application Rates | Ref. |
|---|---|---|---|---|---|---|---|---|
| Lab | Agricultural residues | 450 | Loamy | Slightly decreased | Slightly increased | Increased | 0.46% (W/W) | [37] |
| | | | Sandy | Slightly decreased | Slightly increased | Increased | | |
| Lab | Woody residues | 620 | Sandy loam | Decreased | Increased | N/A | 1%, 5%, 10% and 20% (V/V) | [38] |
| | | | Sandy | Decreased | Increased | | | |
| Field | N/A | N/A | N/A | Decreased | Increased | Increased | 5 and 10 tons ha$^{-1}$ | [39] |
| Field | Sewage sludge | 700–850 | Loamy sand | Decreased | N/A | N/A | 20, 40, and 60 tons ha$^{-1}$ | [40] |
| Lab | Agricultural residues | 450 | Sandy loam | Decreased | No effect | Increased | 27.5 tons ha$^{-1}$ | [41] |
| | | | Clay loam | Decreased | Increased | Increased | | |
| Field | Agricultural residues | 200–600 | Loam sand | Decreased | N/A | Increased | 10, 25, and 50 tons ha$^{-1}$ | [42] |
| Field | Agricultural residues | 360 | Sandy loam | Decreased | N/A | Increased | 4.5 and 9 tons ha$^{-1}$ | [43] |
| Lab | Agricultural residues | 300 and 700 | Desert | Decreased | Increased | Increased | 5% (W/W) | [44] |
| Field | Woody residues | N/A | N/A | Decreased | Increased | Increased | 55 tons ha$^{-1}$ | [45] |
| Lab | Agricultural residues | 350–650 | Sandy | N/A | Increased | N/A | 1%, 2%, 3%, and 4% (W/W) | [46] |
| Lab | Woody residues | 350 | Sandy loam | Decreased | Increased | Increased | 2%, 4%, and 6% (W/W) | [47] |
| Field | Woody residues | 500 | Silt loam | Decreased | Increased | Increased | 24 and 46 tons ha$^{-1}$ | [48] |
| Field | Agricultural residues | 550–600 | Clay loam | Decreased | N/A | Increased | 10, 20, and 30 tons ha$^{-1}$ | [49] |
| Yard | Agricultural residues | 400–450 | Planosol | Decreased | Increased | Increased | 1%, 2%, and 3% (W/W) | [50] |
| Field | Agricultural residues | 550 | Haplic Luvisol | Decreased | Increased | Increased | 10 and 20 tons ha$^{-1}$ | [51] |
| Lab | Forest residue | 450 | Desert sandy | Decreased | Increased | Increased | 39.5, 58.7, and 65 tons ha$^{-1}$ | [52] |
| Field | Agricultural residues | 550 | Sandy clay loam | No difference | Decreased | No difference | 5.5, 16.5, and 33 tons ha$^{-1}$ | [36] |
| Field | Woody residues | 580 | Luvisol | Decreased | Increased | Increased | 25- and 50-tons ha$^{-1}$ | [53] |

A most recent meta-analysis showed an average increase in soil porosity by 6.27%, decrease in bulk density by 7.47%, and increase in water-holding capacity by 9.82% [54]. However, biochar derived from softwood and walnut shell did not affect soil porosity or water retention over a period of six years in silty clay soil. It was suggested that the effect on soil porosity and water retention was temporary until the pores of biochar were occluded with clay or soil organic matter (SOM) [55]. Woody biomass derived-biochar was shown to have no effect on soil porosity or water retention after four years of amendment [56].

Comparably, in the case of soil bulk density, a majority of the studies reported a decrease after BC amendment [34,35,54]. There were also a few reports where no significant decrease in bulk density was observed [57–60]. A majority of the data from previous meta-analyses and reports indicate that the addition of BC to a coarse, textured soil had a larger positive impact on soil physical properties compared to clay textured soil [34,35,54,61,62].

Biochar produced from wheat straw (550–600 °C) incorporated with clay loamy soil improved its physical properties and enhanced the yield of wheat when irrigated with saline water [49]. The biochar amendment decreased soil bulk density by 5.5–11.6% and increased porosity by 35.4–49.5%. The biochar amendment also seemed to mitigate soil sodicity and also increased total NPK (nitrogen, phosphorous, potassium) availability in mixed soil layer. This resulted in the improvement of wheat yield by 8.6% and 8.4% at the BC application rates of 10 and 20 t/ha, respectively [49]. However, at the application rate of 30 t/ha$^{-1}$, the improvement in yield was the lowest (2.2%), which was probably due to high salinity and the immobilization of N. This study suggests that for saline irrigation in clay loamy soil, the optimal application rate of BC produced from agricultural residue should be between 10 and 20 tha$^{-1}$ [49]. These studies imply that initial soil characteristics, along with BC application rate and type, determine the final changes in the physical properties of the soil.

### 2.3. Soil–Water Relations

Accessible fresh water supplies are becoming increasingly limited, and 70% of available fresh water supports crop irrigation [63]. Although biochar holds promise for improved hydrological functions, there are differing schools of thought regarding the role of BC in improving the long- term water-holding capacity of soil [64]. BC amendment has been reported to increase rainfall absorption and soil water-holding capacity, particularly in non-irrigated production regions [65–67]. However, the pre-existing physical and biochemical characteristics of the soil and the wide array of BC production parameters (feedstock inputs, pyrolysis temperatures, application methods, and geographical variables) ultimately determine the BC's impact on water-holding capacity. In order to probe the influence of BC on water dynamics, initial experiments were performed with soil columns in greenhouses with the addition of farm or potting soils. Field studies are now becoming prevalent in peer-reviewed literature, particularly within the last 10 years.

The identification of key features that contribute to improved water retention could lead to an expanded role for BC in crop production. Overall, it was determined that feedstock selection and pyrolysis temperature, the most predictive variables impacting water status, impact BC surface chemistry and porosity, the latter of which is a major contributor to the water-holding capacity of BC [68,69]. Pore saturation is highly dependent on BC surface chemistry, which is affected by pyrolysis temperature. An increase in pyrolysis temperature volatilizes organic elements and thermally cracks the biomass, thereby rendering hydrophobic compounds more hydrophilic and increasing the overall BC porosity [70]. Conversely, BCs produced via low-temperature pyrolysis exhibit negative capillary pressure, inhibiting the hydration of the pore space [71].

Comparative analysis of Fourier transform infrared (FTIR) spectroscopy data collected from nine different feedstocks pyrolyzed at 250 °C, 500 °C, and 700 °C revealed the relationship between BC surface chemistry and hydrophobicity [72]. The spectrometer data indicated that the functional group C=O in carboxylic acid was present only in the BC obtained from pyrolysis at 250 °C, making it hydrophobic. BC produced at 500 °C and 700 °C were deemed more appropriate for improving soil

water status. A significant correlation was identified between low pyrolysis temperature (<300 °C) and surface functional groups (specifically acidic moieties), and increased hydrophobicity contributing to low water retention was reported [68,73]. Other factors, including cation exchange capacity, play a role along with the variables of surface groups and porosity in determining the hydrophobic properties for each specific BC [74].

Considerable variation in total pore volumes was reported in BCs produced at 400 °C, 600 °C, and 800 °C from various feedstocks. Wood-based BC possessed a comparatively higher range of micropores (5–30 μm), and although the number of micropores decreased with increasing pyrolysis temperatures, this BC still retained relatively large pore volumes overall due to pyrogenic micropores. In contrast, the pore volumes of BC derived from poultry manure and agricultural wastewater sludge were smaller, indicating that these feedstocks may not be suitable for improving water retention in amended soils [75]. While BC amendment imparts large increases in porosity, permeability, and moisture retention in clay soils, these affects are diminished in silt loam soils [76,77]. The particle size of BC had a clear impact on soil bulk density, with a linear decrease in bulk density of sandy soil observed when large-particle-containing hardwood BC (620 °C) was used. Smaller BC particles increased water-holding capacity compared to larger BC particles [38], which was possibly due to the increased microporosity resulting from higher pyrolysis temperatures. Despite this, the addition of BC at 25 Mg ha$^{-1}$ to sandy soils did not result in increased water retention. In a study with *Miscanthus giganteus* residue-derived BC (450 °C), the increased porosity of larger BC particles proved beneficial for soil water retention, while smaller BC particles under 0.15 mm retained water too well, thereby strongly reducing its bioavailability [37].

BC was reported to increase the water-holding capacity in coarse and medium textured soils by an average of 51% and 13%, respectively [78]. This was attributed to a higher abundance of soil micropores resulting from the intrinsic microporosity of BC. However, a reduction in water-holding capacity was reported in fine-textured soils, which was possibly due to the overall decrease in micropores or occlusion of existing pores. Field studies of high-porosity BCs derived from softwood (600–700 °C) and walnut shell (900 °C) reported a temporary improvement of water-holding capacity; however, no long-term improvement was seen in BC-amended silty clay loam soils subjected to a corn–tomato rotation with conventional or organic production regimes [55]. Plant-available water in fine-textured soils could be enhanced through the management or manipulation of hydrophobic properties of BC, thereby improving BC–soil interactions [78]. For example, it has been reported that grapevine feedstocks subjected to low pyrolysis temperatures (approximately 400 °C) yield BC with a 23% higher available water content in clay soils [79].

## 2.4. Soil Tilth and Nutrient Status

Defining management approaches to increase the productivity of agricultural soils remains a priority as food demand increases and arable farmland decreases [80]. As a mineral-rich organic material, BC can be incorporated into agricultural soils, potentially serving as a slow-releasing fertilizer, positively affecting soil tilth and enhancing the nutrient status of agricultural soils [81–83]. The basis for this potential use lies in the unique porosity of BC, its facilitation of chemical and physical interactions between nutrients and the carbon material, and its strong intrinsic sorption properties. Due to the large surface area, porous microstructures, and negative surface charge, BC enhances nutrient retention in the soil. Furthermore, the nutrient retention properties of BC may significantly reduce irrigation or the rainfall-induced leaching of water-soluble minerals [66,84]. The slow desorption of the BC-sequestered nutrient elements may supply a steady rate of nutrient delivery, thereby alleviating the need for excessive fertilizer use. Together, these agronomic benefits to soil health may also mitigate freshwater eutrophication that results from fertilizer runoff, prevent pesticide contamination, and reduce the risk of environmental damage [85–87].

While composition varies based on feedstock and pyrolysis parameters, a universal characteristic of BC is that it is carbon-dense, which facilitates the retention of necessary plant nutrients such as N, P,

K, Mg, Fe, and Ca [88–91]. Depending on the soil status and existing nutrient deficiencies, BCs can be custom-manufactured to replenish depleted nutrients. It has been demonstrated that BCs derived from different feedstocks possess variable amounts of beneficial plant nutrients [47,66,91–94]. The general characteristics of three major BC feedstock sources are as follows:

- Organic waste feedstocks, such as animal manure and sewage sludge-derived BC, are rich in potassium and phosphorus, low in C levels, and low in surface area; additionally, eggshell-derived BC is elevated in calcium levels
- Wood-based BC is high in organic matter and surface area, while low in CEC and N, P, and K levels
- Crop residue-derived BC properties reside somewhere in between those of the two previous categories, with specific crops producing BC with different properties (e.g., wheat and rice BC is high in silicon content; soybean BC is high in N).

These feedstocks can be blended in appropriate ratios to produce BCs with desired nutrient and/or mineral profiles. Further modifications, including the alteration of pyrolysis parameters, physical alterations, chemical modifications, and BC-mediated composting have been discussed to aid in the customization of BC to ameliorate detrimental soil aspects [91].

Soil pH and the abundance/availability of important plant nutrients such as phosphorus (P) and nitrogen (N) are positively affected in BC-amended soils. Limiting the pyrolysis temperatures to less than 700 °C enhances the levels of P and N in BC, both of which can be lost at higher temperatures due to the volatilization and transformation of $NH_4^+$ to heterocyclic-N [92]. Wood-derived BC (450–550 °C) applied at 20 t/ha significantly improved the bioavailability of P in sandy soils, which is an effect that was primarily attributed to the perturbation of abiotic processes (adsorption/desorption of P, altered redox potentials, development of organomineral aggregations) [95]. BC nitrogen levels are correlated closely with the original source of the char; feedstocks high protein biomass, such as grasses, generate BCs with higher N levels (approximately 10% by weight), while wood-derived BCs tend to be N-poor (approximately 1% by weight) [92]. While several individual studies show that wheat BC (450 °C) applications increased total soil N [96], other studies found significant decreases of soil $NH_4^+$ and $NO_3^-$ following BC addition. The latter outcome was likely due to the inherent recalcitrance of the small amount of extractable inorganic N and organic N present. In the studies reporting increased N, this increase could be attributed to a heightened abundance of recruited microorganisms, which assist in the degradation of soil organic nitrogen [97].

In addition to improving mineral nutrient retention, BC has a role in the amelioration of soil erosion and the improvement of overall soil structure [98,99]. A study utilizing hardwood (600 °C) BC at 15 and 30 t/ ha concentrations to amend clay-rich soils in incubation containers demonstrated improved soil aggregate structure and soil stabilization [100]. This is likely due to the interaction of carboxylic and phenolic functional groups on the BC surface, resulting in the formation of cation bridges and consequent BC–mineral complexes [101]. For example, microaggregates observed to form upon the incorporation of hardwood-derived BC (700 °C) into soil with application rates of 2.5% or 5% correlated with a 50–64% decrease in soil loss, respectively [102]. An additional study with oak wood-derived BC applied at a rate of 10 Mg/ha provided further evidence for the stabilizing effects of BC, with significant decreases in soil loss of almost 20% observed in a simulated rainfall experiment. In addition to improving soil retention, BC appeared to reduce the impact force from rainfall, thereby facilitating the reduction of particle detachment [103].

## 2.5. Soil Acidification

The expanding global incidence of soil acidification is concerning, with acidic soils (pH < 5.5) currently accounting for approximately 50% of arable land [104,105]. The excessively low pH of acidic soil results in reduced productivity and decreased crop fertility. The main causes of soil acidification include the use of ammonia-based fertilizers and low nitrogen-use efficiency. In soil, ammonia fertilizers are converted to nitrates and hydrogen ions. The hydrogen ions that are left

over following the uptake of nitrates by crops or after nitrate leaching increase the soil acidity [106]. The removal of crop residue also accelerates soil acidification. An excessive reduction of pH leads to the increased solubility of soil-bound aluminum; thus, soil acidification generally leads to aluminum (Al) toxicity [105]. Aluminum toxicity, in turn, leads to deficiencies in phosphorus, calcium, magnesium, and potassium cations and contributes to impaired root growth.

Current strategies to alleviate soil acidification include liming, the application of crop residue, and the use of industrial products; however, these methods have several disadvantages. For example, liming material elicits a disproportionately strong effect on top surface soils in comparison with lower layers. This method is also costly, due to the high transportation costs of liming material [106,107]. The application of industrial products can lead to heavy metal toxicity [108]. Similarly, the excessive application of organic material may lead to both heavy metal accumulation and eutrophication, the latter resulting from augmented concentrations of nitrogen and phosphorus [26,109]. Hence, biochar, which is naturally alkaline, is a potential solution to the problem of soil acidity.

Various studies have validated the effectiveness of BC in reducing soil acidity [110–112], and a linear correlation of biochar alkalinity with the resulting soil pH has been established [105]. Furthermore, it has been demonstrated that the increased pH-buffering capacity of BC-amended soils is due to a BC-derived increase in cation exchange capacity [113]. The carbonates and oxides of cations such as Ca, K, Mg, Na, and Si formed during pyrolysis are known to react with dissolved Al and hydrogen ions in soil, leading to increased soil pH and decreased Al uptake by the plants [114]. Previous meta-analyses and individual studies have concluded that in imparting increased buffering capacity, BC amendment can increase the soil pH by >2.0 units [115]. Not surprisingly, the original feedstock material plays a key role in determining the final pH of BC. For example, BC generated from manure has higher alkalinity, pH-buffering capacity, and propensity for the alleviation of Al toxicity, compared to crop residue-derived BC. Thus, the former would be more suitable for extremely acidic soils [116]. Soils exhibiting Al toxicity could be reclaimed via BC amendment, the ash content of which would precipitate $Al_3^+$ to less toxic $Al(OH)_3$ and $Al(OH)_4$ [117]. Furthermore, carboxyl and other organic functional groups on the BC surface would provide additional sites for $Al_3^+$ binding [117]. Functional groups such as $COO^-$ and $O^-$ also contribute to the alkalinity of biochar through reaction with free $H^+$ ions [27,118].

In soil, $H^+$ is produced through the aerobic conversion of ammonia to nitrate. Experimental results have demonstrated that BC amendment leads to decreased soil nitrification through the adsorption of $NH_3$ and $NH_4$ onto the BC surface. Soil amendment with wheat straw-derived BC (500 °C) led to reduced nitrification in cadmium-contaminated Ferralsol soil by decreasing soil acidity (Table 2) [119]. Similarly, amendment with pig manure-derived BC (300 °C) resulted in decreased soil acidification and increased cation exchange capacity [120,121], and crop residue-derived BC (500 °C) led to improved rice growth, yield, and soil nutrient availability in acidified soil [122]. Collectively, information from the literature has established that carbon content, nutrient availability, and alkalinity are highest when BC is generated from manure feedstock, intermediate when generated from crop residue feedstock, and lowest when generated from woody plants-based feedstock (Table 2). Finally, biochar produced at higher temperature has higher pH and might be more suitable for countering soil acidity.

**Table 2.** Selected recent studies documenting the impact of biochar on acidified soils.

| Exp. Type | Soil Type | BC Feedstock | Pyrolysis Temperature (°C) | Effect of Biochar Amendment | BC Application Rates | Reference |
|---|---|---|---|---|---|---|
| Lab | Ultisols | Crop residue | 400 | Inhibited soil re-acidification and increased pH buffering capacity | 3% (W/W) | [105] |
| Lab | Ferrosol | Crop residue | 450 | Promoted nitrification and inhibited re-acidification of Cd-contaminated soils | 3% (W/W) | [119] |
| Lab | Sandy | Pig manure and poultry litter | 300 | Decreased soil acidification and increased cation exchange capacity | 0.5 %, 1%, and 2% (W/W) | [120] |
| Lab | Ultisols Oxisol | Crop residue | 400 | Increased soil pH buffering capacity and increased the resistance of soils to re-acidification | 3% and 5% (W/W) | [123] |
| Lab | Ultisols | Crop residue | 400 | Increased soil pH, neutralized soil acidity, increased soil pH-buffering capacity, and increased resistance of soils to re-acidification | 1% and 3% (W/W) | [124] |
| Lab | N/A | Crop residue | 500 | Biochar significantly promoted rice growth and the yield increased in acidified soil | 2% (W/W) | [122] |
| Lab | Oxisols | Crop residue | N/A | Alleviated soil acidification | 1%, 2% and 5% (W/W) | [125] |
| Lab | Loamy sand | Sewage sludge | 300 | Reduced soil acidification | 0.5%, 1% and 2% (W/W) | [126] |

## 3. Biochemical Properties

### 3.1. Soil Organic Matter (SOM)

Soil organic matter comprises the total organic carbon in a soil and is the main determinant of overall soil fertility. SOM components consist of plant residue, animal waste, microbial populations, and active and stable organic matter in soil. SOM contributes to soil fertility by serving as a nutrient source for crops and microbes, causes soil aggregation, and improves water retention and nutrient exchange. It also helps to reduce soil compaction and surface crusting. It has been reported that the impact of biochar on SOM depends on the following variables [127–129]:

1. Type of biomass used for production of BC
2. Pyrolysis temperature
3. Pre-existing SOM levels in the soil

Amending soils with biochar often results in alterations in C cycling and mineralization, and this effect is known as 'priming'. Previous studies have reported both positive and negative effects of priming. Grass-derived BC produced at lower temperatures (250 °C and 400 °C) resulted in positive priming resulting in increased C mineralization. However, BC produced at higher temperatures (525 °C and 600 °C) from hardwood resulted in negative priming [128]. It was hypothesized that negative priming resulted from the organic matter binding to the biochar and thereby becoming unavailable to microbial and enzymatic action.

Analogous results were observed in a study that showed that crop residue-derived biochar produced at lower temperature (300–550 °C) generally resulted in positive priming when applied to arable and fallow soils; however, in the case of grassland soils, the effect was negative [127]. In another study, which aimed to analyze the short-term effect of biochar on SOM, BC produced from woody feedstocks at lower temperatures (350 °C) had greater positive priming during 0–13 days of biochar application both in low and high pH clay loam soil [130]. The extent of positive priming was reduced for low and high pH clay loam soil when BC produced at higher temperature (700 °C) was used. The addition of fresh labile substrate, such as rye grass, to BC produced at both high and low temperatures further increased priming and mineralization [130]. If the goal is to sequester carbon, rapid mineralization caused in conjunction with low temperature-derived BC results in carbon loss, necessitating BC reapplication. The application of high temperature-derived BC can also be used to reduce the priming effect and aide in carbon sequestration.

The addition of 3% (w/w) BC prepared from forest residues at 550 °C has been reported to delay the decomposition of SOM and reduce N mineralization when added to acidic red loam soil [131]. However, some studies did not find BC to contribute to SOM decomposition [132,133]. A reduction in priming and a 16% reduction of SOM decomposition was reported when crop residue-derived BC pyrolyzed via gasification at 1200 °C was added to sandy loam soil [134]. The reduction may be due to a shift in the preference of the microbial community for biochar as a C source [135]. There have been various studies in which BC produced at lower temperature ranging from 450 to 550 °C stimulated positive priming when added to sandy loam (Table 3). Overall, BC promoted increases in C sequestration, organic carbon retention, SOM, mineralization, phosphorous and potassium content, and plant biomass [136–139]. Conversely, one study reported a decrease in soil microbial biomass and SOM mineralization when crop residue-derived BC (450 °C) was applied to sandy loam soil [140].

**Table 3.** Selected recent studies summarizing the effect of biochar on soil organic matter (SOM).

| Exp. Type | Soil Type | BC Feedstock | Pyrolysis Temperature (°C) | Effect of Biochar Amendment | BC Application Rates | Reference |
|---|---|---|---|---|---|---|
| Lab | Acidic red loam | Forest residue | 550 | Decomposition of SOC(soil organic carbon) declined and reduced mineralization of SOM | 1% and 3% (W/W) | [131] |
| Field | Sandy loam | Crop residue | Gasification at 1200 | Reduced SOM degradation by 16%. | 30 tons ha$^{-1}$ | [134] |
| Lab | Podzol Antric | Woody biomass | 550 | Increased the SOM mineralization | 1% (W/W) | [141] |
| Field | Sandy | Crop residue | 350 | Increased soil organic matter and N | 5% (W/W) | [142] |
| | | | 450 | Decreased organic matter and N content | | |
| Field | N/A | Sewage sludge biochar | N/A | SOM are increased | 16.5 tons ha$^{-1}$ | [136] |
| Field | Plaggic Anthrosols | Crop residue | 350 | Positive priming | N/A | [137] |
| Field | Silt loams | Woody biomass | 900 | Increased soil organic matter, soil pH, phosphorus, potassium, sulfur, and the shoot and root biomass of wheat | 12, 24.6, and 49.3 tons ha$^{-1}$ | [138] |
| Field | Sandy loam | Crop residue | 450 to 500 | Decrease of SOM mineralization, reduce soil microbial biomass | 5.5 tons ha$^{-1}$ | [140] |
| Lab | Sandy loam | Woody biomass | 450 | Increased organic carbon retention and promoted carbon sequestration | 2%, 5%, and 10% (W/W) | [139] |
| Field | Sandy loam | Crop residue | 360 | SOC increased after biochar application and did not contribute to soil aggregation | 4.5 and 9 tons ha$^{-1}$ year$^{-1}$ | [43] |
| Lab | Sandy loam | Crop residue | 600 | Significantly increased SOM, microbial respiration, and microbial biomass | 0.5% and 1% (W/W) | [143] |

*3.2. Microbial Activity*

Considerable emphasis has been placed on the topic of microbial dynamics in agricultural systems and their role in crop productivity. The health and diversity of soil microbial populations as a function of agro-ecosystem well-being has diverse implications for water-use efficiency, soil structure and stability, nutrient cycling, disease resistance, and eventual crop productivity [144,145]. While other organic amendments are only stable for relatively short periods in the soil environment, BC is more stable and remains in the soil for hundreds to thousands of years, as it is not easily degraded, and it could support soil microbial communities for an extended period of time with reduced inputs [66].

The diverse and specific physiochemical characteristics of BC that influence soil microbial composition are increased labile carbon, pH, surface area for colonization, and water content in amended soils. BC addition induces remodeling of the microbial diversity and community structure of the soil; however, the changes are highly variable and dependent on the individual soil properties [146,147]. It was reported that low pyrolysis temperature BCs (>350 °C) harbor a greater number of organic residues and are commonly characterized by lower pH. In contrast, at high temperatures (<600 °C), the abundance of organic moieties contributes to the production of a higher pH BC. It was concluded that pyrolysis temperature (and the BC-related characteristics associated with temperature) is the single most important factor that determines how the microbial communities are influenced [148]. Overall, there is a consensus that BCs foster the growth and maintenance of soil microbial communities [95,149–151].

### 3.2.1. Fungi

In terms of their abundance and diversity, both beneficial [152–154] and detrimental [95,155–157] effects of BC on fungal communities have been reported. In comparison to bacteria, fungi respond differently to organic and inorganic treatments. Soil bacteria act as better indicators of soil fertility than soil fungi [158]. The mechanisms for improved fungal diversity and abundance appear to be correlated more with the physical microstructure of BC and the recalcitrant organic carbon than other factors. This was demonstrated in a study where corn straw BC (500 °C) derived aqueous extractable substances and organic extractable substances, and the remaining solid BC were tested [159]. It has been hypothesized that BC addition preferably fosters bacterial communities over fungal communities. The bacteria may starve the fungi of C and therefore outcompete them [155].

It was demonstrated that fungal diversity was lowered in soybean and rice straw BC (500 °C) soils compared to controls, although individual order, family, genus, and species level fungal communities were affected differently [160]. These outcomes could be a result of the "unbalanced competition" theory. This theory describes the phenomena of saprotrophs exponentially increasing their abundance due to the easily mineralizable carbon found in BC, therefore leading to an overall decrease of other fungal groups and potentially suppressing their abundance and diversity [161]. Other speculations underlying decreased fungal diversity and population include the high levels of organic compounds, mineral elements, and higher soil pH due to BC amendment [162].

### 3.2.2. Bacteria

The microbial community consisting of bacteria tend to respond positively to BC, as several studies have reported a significant increase in abundance and diversity, after BC application, especially in the rhizosphere soil [162–165]. For example, an increase in specific bacterial families and species such as phosphorous solubilizing [166], nitrifiers [167], and N-fixing and denitrifiers [168] was reported with *Malus pumila* woodchip BC (500 °C), *Eucalyptus saligna* hardwood BC (550 °C), and sugar maple wood BC (400 °C) soil amendment, respectively. Additional studies found only modest or no differences [140,159,168,169]. The change in the composition of bacterial communities after the incorporation of BC in soil is highly dependent on the pre-existing bacterial community, soil type, and overall BC characteristics.



Generally, the Gram-negative bacterial community is favored in the nutrient-enriched BC-amended soils and initially predominates the soil environment since it performs specific and narrow functions. They outcompete Gram-positive bacteria that rely on recalcitrant C as their main energy source. Gram-positive bacteria become the dominant bacteria type over time due to BC's ability to form stable aggregates with soil organic matter (SOM) [148]. Utilizing sugarcane-derived BC (450 °C), it was found that bacterial populations increased significantly while fungal populations were significantly reduced in heavy metal-contaminated soils. This was possibly due to the enhanced heavy metal immobilization by the BC addition, although other factors may have contributed to the observations [162]. Similar results with wood (fir, cedar) BC (450–550 °C) indicated a significant shift toward a bacteria-dominated microbial community in a short-term study (3 months) and was attributed to the increased release of labile C from the BC or stable SOM–BC aggregates [95]. A study utilizing bamboo BC (500 °C) provided further support for the concept that bacteria are more sensitive to BC compared to the fungal community, which was mostly due to increased pH with increasing BC addition [170]. These results indicate that alkaline conditions due to BC amendment (liming effect) favor and promote bacterial growth and may inhibit fungal growth.

The high complexity of BC–soil interactions and microbial community dynamics leaves many 'gray areas' in this field that require further investigation. However, assessing the long-term effects of BC-amended soils and microbial population diversity and activity are highly recommended for future studies. Research has indicated that the BC surface and pores can be inundated with plant exudates and dead cells, inorganic and organic complexes, and larger soil microorganisms. These factors may reduce the total available space for microbial colonization of aged BCs over time [171].

### 3.3. Abiotic and Biotic Stressors

### 3.3.1. Heavy Metals

Soil contamination with organic and inorganic toxins increases environmental and agricultural risks and poses a threat to both plants and humans. Efforts to develop remediation processes that bind the contaminants, limit their mobility and bioavailability, and foster improved soil health are ongoing. Currently, organic materials such as charcoal, soot, kerogen and activated carbon are used as amendments for limiting and reducing the bioavailability of multiple soil contaminants [172,173]. The organic contaminants have been shown to sorb preferentially to the carbonaceous fractions present in soil, limiting their bioavailability [174]. BC has also been shown to reduce the bioavailability of heavy metal contaminants. Several studies analyzed the effect of BC amendment on soils contaminated with heavy metals such as arsenic, cadmium (Cd), copper (Cu), nickel (Ni), lead (Pb), and zinc (Zn) [175–177]. The high surface area of BCs results in more effective contaminant binding; however, one of the recent meta-analyses pointed out that the pyrolysis temperature at which a given BC is produced influences its remediation efficiency and type of contaminant that can be removed [175]. A majority of studies tested the effect of BC on Cd pollution and concluded that the higher BC surface area had a smaller effect on Cd bioavailability [175]. The BCs produced at lower temperatures (300–500 °C) have a higher density of functional groups, while BC produced at higher temperatures results in a larger surface area and lower density of functional groups. Another study revealed that BC produced from wheat straw at 450 °C, with a higher density of functional groups, was more effective in treating Cd and Pd-contaminated soil [178]. However, BC produced at higher temperature is also more alkaline and results in the immobilization of heavy metals in acidic soil via the liming effect. The addition of rice and wheat straw-derived BC in soils contaminated with Pb, Cu, Cd, and Zn led to a reduced mobility and bioavailability of the heavy metals, resulting in increased yields and a decreased enrichment of heavy metals in the tested plants [176]. Biochar is also used for the remediation of soil from contaminated sites due to rapid industrialization. It has been recently demonstrated that BC derived from pine wood was able to reduce the bioavailability of Cd, Pb, and Zn in metalloid-contaminated soils at a smelting

site and promoted plant growth [179]. Biochar derived from hardwood (600 °C) was shown to be effective in reducing Ni and Zn by 83–93% in a historically polluted site in the United Kingdom [179].

The effect of sewage sludge BC pyrolyzed at 330–500 °C on alleviating heavy metal toxicity was evaluated. It was observed that with the increase in pyrolysis temperature, the availability of heavy metal in tropical soils was decreased [180]. This might be due to the increased pH, pore size, and surface area in the BC produced at higher temperatures leading to the formation of carbonates, sulfates, phosphates, and metal hydroxides [180]. Due to the reduction in the bioavailability of heavy metal cations, maize yields increased in BC-amended soil in comparison to NPK-fertilized soil [180]. Irrigation with untreated wastewater leads to the accumulation of lead, cadmium, zinc, and iron, which can be taken up by plants or leach into ground water, adversely affecting plant growth and human health. In a recent study, the effect of plantain peel-derived biochar (450–500 °C) on potato yield was studied in sandy soil irrigated with wastewater [181]. This BC regime resulted in the adsorption of soil Cd and Zn and the reduction of the Cd level by 69% and 33% in tuber flesh [181].

A summary of selected studies reporting the effect of BC on the alleviation of heavy metal toxicity is presented in Table 4. Most of the listed studies were carried out in pots. Therefore, large-scale field studies are required to understand the interactions among a particular biochar, soil type, and contaminant. To use biochar for soil remediation, the specific soil and biochar properties must be taken into consideration. Some studies show that certain BC amendment results in high heavy metal immobilization. However, if the mechanism of immobilization is only physical adsorption or cation exchange, these BCs may not be suitable for long-term remediation due to weakly bound metals. BCs that immobilize heavy metals through precipitation or complex formation should be used for long-term remediation.

**Table 4.** Selected recent studies reporting the effect of biochars (BCs) derived from various feedstocks on heavy metal remediation in different types of soils.

| Exp. Type | Soil Type | BC Feedstock | Pyrolysis Temperature (°C) | Effect of Biochar Amendment | BC Application Rates | Reference |
|---|---|---|---|---|---|---|
| Lab | N/A | Wood, bamboo, rice straw, and walnut shell | 500 | Reduced Zn, Cd, Cu, and Pb solubility | 5% (W/W) | [182] |
| Lab | Aridisols | Woodchip-derived biochar | 300 | Reduced extractable Cd, Pd, Ni, and Cu. Improved antioxidant enzyme activity. Increased rapeseed fresh shoot biomass, fresh root biomass, total chlorophyll, total pigments, carotenoids, and lycopene concentration | 1% and 2% (W/W) | [183] |
| Lab | Sandy loam soil | Wood derived biochar | 350–500 | Reduction in the accumulation of Cu and Zn in spinach | 5% and 10% (W/W) | [184] |
| Lab | N/A | Switchgrass and poultry litter | 700 | Decreased the Zn, Cd, and Pb bio-accessibility | 0.5%, 1.0%, 2.0%, and 4.0% (W/W) | [185] |
| Lab | Paddy soil | Wheat straw | 450 | Reduced soil Cd bioavailability | 5% and 15% (W/W) | [186] |
| Lab | Clay soil | Corncob biochar | 600 | Reduced lead leaching | 5% (W/W) | [187] |
| Lab | N/A | Wheat straw | 350–650 | Lower temperature BC led to increased Zn (II) and Cd (II) immobilization acidic condition, and higher temperature BC led to increased Zn (II) and Cd (II) immobilization alkaline condition | N/A | [188] |
| Lab | N/A | Manure | 500 | Promoted Zn and Cd precipitation and reduced total Cd and Zn concentrations in switchgrass shoots and roots | 0%, 2.5%, and 5%, (W/W) | [189] |
| | | Poultry litter | 500 | | | |
| | | Lodgepole pine | 500–700 | Reduced Zn concentration in roots | | |
| Lab | N/A | Rice husk biochar | 550 | Decreased leaching of Cd, Cu, Pb, and Zn | 0.5%, 1%, and 2% (W/W) | [190] |
| | | Maple leaf biochar | | | | |
| Lab | Stagnic Phaeozem | Pine wood | N/A | Decreased heavy metal accumulation in above-ground parts of *Hordeum vulgare* | 2.5% (W/W) | [191] |

### 3.3.2. Salt

Salt stress is known to negatively affect soil properties, plant development, and crop productivity due to disturbed soil structure, soil organic matter, microbial activity, and C:N ratio. Salt stress causes oxidative stress in plants, down-regulating antioxidant enzyme activity [192]. Due to excessive salinization, the sodium ions bind to cation exchange sites in soil, causing poor crop growth and yields. Although saline soil can be reclaimed by washing or excessively irrigating with water to remove excessive salts, it is neither economically nor physically feasible for large fields [193]. On the other hand, sodic soils require treatment with other cations such as calcium to remove excess sodium from cation exchange sites followed by the leaching of sodium [193]. The application of organic amendments such as manure or compost has been shown to improve soil fertility by reducing salt stress. In saline soil, the organic amendment improves soil porosity, leading to the leaching of excess salt. In sodic soil, organic amendments might help by improving the physical characteristics of soil, such as triggering cation exchange with the calcium present in organic amendment and Na present in soil.

Several studies have reported the positive impact of BC on the nutrient status, conductivity, and improved physical and chemical properties of soil. The variable amount of plant nutrients present in the BC can compensate for the nutrient deficiency and improve the fertility of saline soils. For example, sodification raises soil pH, thereby limiting the bioavailability of P. In such soils, BC can act as a P source and improve its availability, aiding plant growth [194].

It has been demonstrated that the mixture of hardwood and softwood biochar produced at 500 °C, when mixed with sandy loam soil and irrigated with a saline solution, improved the yield of potato, maize, and wheat [195–197]. It was also shown that BC was able to reduce the $Na^+/K^+$ ratio in the xylem sap of wheat and potato and reduced Na concentration in maize xylem [197]. When BC produced from wheat straw (350–550 °C) was combined with poultry manure and incorporated into Aqui-Entisol soils, a decrease in Na uptake was observed, leading to increased biomass in maize and an increase of yield in wheat [194,198]. Similarly, rice straw-derived BC (600 °C) alleviated salt stress in paddy soil. There was a significant reduction in bulk density, electrical conductivity, exchangeable Na, and exchangeable chlorine ions in the soil, creating favorable conditions for rice seedling growth [199]. A selection of recent reports in which BC addition was reported to alleviate salt stress is summarized in Table 5.

A majority of the reports support the role of BC in improving soil health, plant growth, and the biological properties of soil. It has been reported that BC adsorbs Na salt and improves plant growth; however, salt-affected land is only considered reclaimed if the Na salts are removed. Therefore, the repeated application of BC might have negative consequences in a case where an increased accumulation of Na-salt bound BC aggravates the salinization problem [193]. There is a need for a better understanding of how different BCs interact with different types of salt-affected soils prior to the prescription of any recommendations.

**Table 5.** Selected recent studies reporting the effect of biochar on salt stressed soils.

| Exp. Type | Soil Type | BC Feedstock | Pyrolysis Temp. (°C) | Effect of Biochar Amendment | BC Application Rates | References |
|---|---|---|---|---|---|---|
| Lab | Loam clay | Rice straw | 300–600 | Reduced bulk density, electrical conductivity, exchangeable Na$^+$ and Cl$^-$. Reduced salt accumulation in rice seedlings. | 0.3% (W/W) | [199] |
| Field | N/A | Citrus wood | N/A | Improved plant growth and productivity. Improved nutrient concentration in soil, dehydration tolerance, and water retention. | 5 and 10 tons ha$^{-1}$ | [200] |
| Lab | Coastal soil | Wood chips | 600 | Improved photosynthetic performance and alleviated oxidative damage and salt stress. | 5% (W/W) | [201] |
| Lab | Sandy clay loam | Rice straw | 450 | Mitigated oxidative and salt stress. Reduced Cd and Na concentration in plant. | 3% and 5% (W/W) | [202] |
| Lab | N/A | Maple residues | 560 | Improved plant growth and xylem structure. Reduced salinity and plant stress hormones. | 5% and 10% (W/W) | [203] |
| Lab | N/A | Rice straw | 300 | Increased seed germination rates of cowpea. Increased photosynthetic efficiency and photosynthetic pigments. | N/A | [204] |

### 3.3.3. Biotic Stress

Several recent reports have emerged showing BC to aid plants in countering biotic stresses. It has been suggested that BC-mediated nutrient retention, adsorption, pH adjustment, and increased water holding provides plants with the capacity to respond to pathogens and to counter the effect of toxic metabolites generated by plants [205].

The severity of gray mold, powdery mildew, and anthracnose on strawberry plants was evaluated in the presence of 3% (w/w) citrus-derived BC (450 °C). It was observed that greenhouse waste-derived biochar when mixed with coconut fiber/peat reduced the severity of gray mold after disease challenge by 74% in mature strawberry plants and by 53% in young strawberry plantlets. Post-disease challenge, anthracnose severity was reduced by 39–49% and powdery mildew severity was reduced by 68% [206]. Both citrus wood and greenhouse waste-derived BC reduced gray mold severity as well. It was observed that BC application induced the expression of genes related to the systematic acquired resistance and induced systemic pathways, which might have contributed to the reduction in disease severity [206]. The ability of biochar to absorb pathogenic cell wall-degrading enzymes and toxic metabolites produced by soil pathogen *Fusarium oxysporum* was also tested with tomato seedlings [205]. The tomato seedlings were treated with 3% BC produced from eucalyptus wood chips and greenhouse pepper plant waste pyrolyzed at 350 °C and 600 °C. It was observed that seedlings exposed to enzymes from *Fusarium oxysporum* and toxic metabolites without BC developed severe disease-like symptoms, whereas those symptoms were significantly reduced in the seedlings grown with BC amendment [205]. The exact mechanism of interaction with BC is still unclear; however, it was observed that a majority of the fungal enzymes that were immobilized by BC through adsorption were deactivated [205]. A commercial-scale study conducted over a period of 3 years tested the effect of BC on growth and disease resistance in *Capsicum annuum* L. (sweet pepper) [207]. Pepper seedlings were planted in four combinations of sandy soil amended with biochars produced from greenhouse pepper plant waste and eucalyptus chips. During the first year of growth, it was observed that greenhouse waste-derived BC (450 °C) reduced the severity of powdery mildew by almost 50% 168 days post-planting in comparison with controls. In the second year of the study, the greenhouse waste BC (350 °C and 450 °C) showed the highest pepper yield compared to the other treatments and the control [207], in addition to a significant reduction in powdery mildew severity. The incidence of plants affected by broad leaf mite was also reduced when amended with greenhouse waste BC. A comparable trend was observed in the third-year trial. Powdery mildew severity was reduced by 25% in both greenhouse waste and eucalyptus wood

chip-derived BC after 160 days of growth [207]. Biochars produced from greenhouse waste (350 °C) and eucalyptus chips (600 °C) were shown to be effective in suppressing crown and root rot in tomato caused by *Fusarium oxysporum* f. sp. *radicis lycopersici* [208]. The application of greenhouse waste BC at 0.5%, 1%, and 3% reduced disease severity by 72%. The eucalyptus chip BC also reduced disease severity by 44% compared to the control plants [208]. There are also some reports where no significant effect of BC on soil-borne pathogen suppression was observed [209–211].

The number of studies exploring the role of BC in pathogen suppression is significantly less than other organic amendments such as compost, peat, and crop residue. Hence, additional studies are needed in order to understand the mechanism behind the ability of biochar to suppress pathogens and to be able to prescribe biochar regimens as safe and effective amendment strategies for the improvement of plant resistance to soil-borne pathogens.

## 4. Impact of Biochar on Crop Production

Increasing crop yields to feed a burgeoning population is a daunting task in the face of a myriad abiotic and biotic challenges, including the reduction of arable farmlands and increased plant stressors due to the changing climate [212–214]. These issues are especially important in organic production systems where the average crop yields are 5–34% lower compared to conventional farming [215–218]. The use of biochar in soil remediation can be a useful strategy, especially in degraded soils [219,220]. Furthermore, the potential to significantly reduce the organic yield gap through better fertilization regimes has been proposed, suggesting an expanded role for nutrient-rich biochars [221].

A meta-analysis of BC effects on plant productivity concluded that BC use holds promise as a method to increase crop yields and could further promote ecosystem services and carbon storage [91,222–224]. It was noted that increased soil N, P, K, the reduction of soil acidity due to the liming effect of BC, and improved water relations contributed to various soil and crop responses. In this review, a comprehensive literature search was performed in the Google Scholar search engine with the search terms "biochar crop productivity yield" for the years 2017–2019, which yielded 330 entries. These entries were further parsed using minimal criteria terms—BC feedstock source, pyrolysis temp, retention time, and soil type. The second round reduced the number of entries to 18. The results of the literature search are summarized in Table 6.

**Table 6.** Impact of BC on crop productivity summarized from a comprehensive literature search. Soil types listed in the table correspond to the types reported in the original studies. CEC: cation exchange capacity.

| Crop Productivity | | | | Soil type, Experiment Type, Length | Biochar Feedstock | Pyrolysis Temp °C, Residence Time, Application Rate | References |
|---|---|---|---|---|---|---|---|
| **Crop Tested** | **Productivity** | **Beneficial** | **Detrimental** | | | | |
| Cherry tomato (*Solanum lycopersicum*) | Bamboo BC increased tomato yields | Both BCs improved tomato quality with increased total sugars | Rice husk BC did not improve total N % | Clay loamy | Rice husk and bamboo | 500 | [225] |
| | | | | *Greenhouse* | | 1 h | |
| | | | | Short-term ≤ 1 year | | 2% and 5% (w/w) | |
| Lettuce (*Lactuca sativa*) | For both soils, BC rates of 20 and 30 t/ha$^{-1}$ significantly increased above-ground biomass | Effective fertilizer for lettuce production at least for two growing cycles | Biosolid BC could increase harmful soil elements such as heavy metals | Silty loam and sandy loam | Fecal matter | 450 | [226] |
| | | | | *Greenhouse* | | 1 h | |
| | | | | Short-term ≤ 1 year | | 10, 20, and 30 t/ha | |
| Chrysanthemum (*Glebionis coronaria*, cv. 'Crown Daisy') Leaf lettuce | 3% BC significantly decreased yields No effect | BC increased WHC(water holding capacity) and SOM | Higher BC application reduced plant productivity | Pedocals, silt-clay | Peanut shells | 350 | [227] |
| | | | | *Greenhouse* | | 3 h | |
| | | | | Short-term ≤ 1 year | | 0%, 1.5%, 3%, and 5% (w/w) = to 0, 37.5, 75, and 125 t/ha in the field | |
| Beans | Bean yields were significantly reduced with BC application | Increased germination rate in BC-amended soils | Significant decreases in some macro and micronutrients | Krome loamy | *Melaleuca quinquenervia* (Broad-leaved paperbark) hardwood | 350 | [228] |
| | | | | *Greenhouse* | | 7 h | |
| | | | | Short-term ≤ 1 year | | 2% and 5% (w/w) | |
| Wheat (cv. 'Yecora Rojo') | 300 °C BC with NPK increased yields | Increased soil water retention and decreased bulk density | BC alone decreased yields with BC produced at higher temp° (400, 500, 600 °C) | Loamy sand | Date palm tree residues | 300, 400, 500, and 600 | [229] |
| | | | | *Greenhouse* | | 4 h | |
| | | | | Short-term ≤ 1 year | | 8 t/ha | |
| Potatoes (*Solanum tuberosum* L., cv. 'Russet Burbank') | No significant differences in yield | BC increased soil CEC | BC had no effect on leaf greenness rate or photosystem activity | Sandy | Green plantain peels | 450–500 | [230] |
| | | | | *Field Study* | | 18–25 min | |
| | | | | Long-term, 2 years | | 13.5 t/ha (1% w/w) | |
| Tomato & Maize (*Zea mays*) | BC does not have a significant long-term effect on yield | Increased K$^+$, Ca$^{2+}$, and PO4-P in the soil in year 2 | Delayed nutrient availability from BC and short-lived effects | Rincon silty clay loam | Walnut shells | 900 | [231] |
| | | | | *Field Study* | | 1–2 h | |
| | | | | Long-term, 4 years | | 10 t/ha | |
| Winter wheat (cv. 'Xiaoyan no. 22') | Low levels (1%, 2%) of BC had a positive effect on wheat yields | Total nitrogen and SOC increased with BC applications | Under drought conditions, BC addition decreased the availability of nutrients | Silty-clay | Apple wood | 450 | [232] |
| | | | | *Outdoor pot study* | | 8 h | |
| | | | | Short-term ≤ 1 year | | 1%, 2%, 4%, and 6% (w/w) | |
| Maize | BC and fertilizer led to a significant increase in maize yield | BC improved soil water-holding capacity | BC alone had no effect on maize yields | Sandy clay loam | Maize cobs | 500 | [233] |
| | | | | *Field Study* | | 1 h | |
| | | | | Short-term ≤ 1 year | | 20 t/ha | |
| Chinese cabbage (*Brassica rapa*) | BC significantly improved crop yields | BC increased soil pH and CEC | BC did not affect the soil bulk density and porosity | Loamy | Barley straw | 400 | [234] |
| | | | | *Field Study* | | 1 h | |
| | | | | Short-term ≤ 1 year | | 10 t/ha | |

**Table 6.** *Cont.*

| Crop Productivity | | | | Soil type, Experiment Type, Length | Biochar Feedstock | Pyrolysis Temp °C, Residence Time, Application Rate | References |
|---|---|---|---|---|---|---|---|
| **Crop Tested** | **Productivity** | **Beneficial** | **Detrimental** | | | | |
| Radish (*Raphanus sativus* L. cv. French Breakfast) | Increased yields in second year | Reduced bulk density and increased porosity, moisture content, soil pH | No effect on first-year growth | Alfisol or Luvisol | Local hardwoods (*Parkis biglosa*, *Khaya senegalensis*, *Prosopis africana* and *Terminalia glaucescens*) | 580 | [53] |
| | | | | Field Study | | 24 h | |
| | | | | Long-term, 2 years | | 25 and 50 t/ha | |
| Rice (cv. 'Naveen') | Increased grain yield up to 24% | Increased total organic C in soils | Microbial carbon use efficiency decreased due to BC addition | Sandy clay loam | Rice husk | 350 | [235] |
| | | | | Field Study | | 6 h | |
| | | | | Long-term, 3 years | | 0.5, 1, 2, 4, 8, 10 t/ha | |
| Maize (cv. 'hybrid LG 6030') | Increased corn yields | Increased P levels during the two years of cultivation | BC was unable to supply the necessary K for further crop production | Red-Yellow Latosol with clayey texture | Sewage sludge | 300 and 500 | [236] |
| | | | | Field Study | | 30 min | |
| | | | | Long-term, 2 years | | 15 Mg/ha | |
| Okra (*Abelmoschus esculentus* L., cv. 'OH-397') | Increased yields vs. controls | Significant increase in SOC and microbial activity | Lower benefit cost ratios for BC compared to controls | Inceptisol with sandy loam texture | Mixed local hardwoods | 450 | [237] |
| | | | | Field Study | | 4 h | |
| | | | | Long-term, 2 years | | 5 t/ha | |
| Rice (*Oryza sativa* L.) & Wheat (*Triticum* ssp.) | Not affected | BC amendment increased the soil water-holding capacity, soil nutrients, and SOC | Short-term effects and BC alone did not increase yields | Hydragric Anthrosol, sandy | Wheat straw | 350–550 | [166] |
| | | | | Field Study | | 2–3 h | |
| | | | | Long-term, 6 years | | 20 and 40 t/ha | |
| Sunflower (*Helianthus annuus* L., cv. 'Embrapa 122/V2000') | Sunflower seed and oil yield declined | Increased levels of most soil minerals and total carbon levels | Nitrogen levels in leaves and the nitrogen uptake of the entire plant decreased with biochar application | Dark red soil, Typic Hapludalfs | Sugarcane bagasse and sunflower residues | 500–600 | [238] |
| | | | | Field Study | | 1 h | |
| | | | | Short-term ≤ 1 year | | 1% (w/w) | |
| Spring barley (*Hordeum vulgare* L.) | Increased yields with BC + NPK | Increased soil water status in BC amended soils in the first year; increased soil carbon status | BC only decreased yields for both crops compared to control NPK plants | Sandy loamy silt; calcareous Chernozem on loess | Hardwood | 550 | [239] |
| | | | | Field Study | | 2 h | |
| Sunflower | No difference vs. controls | | | Long-term, 2 years | | 72 t/ha | |
| Rice (*Oryza sativa* L.) & Wheat (*Triticum* ssp.) | Not affected | BC amendment increased the soil water-holding capacity, soil nutrients, and SOC | Short-term effects and BC alone did not increase yields | Hydragric Anthrosol, sandy | Wheat straw | 350–550 | [166] |
| | | | | Field Study | | 2–3 h | |
| | | | | Long-term, 6 years | | 20 and 40 t/ha | |
| Cauliflower (*Brassica oleracea*, cv. 'Desire') | No significant improvement in crop yield | No negative effects to crop productivity or soil quality | Soil moisture and bulk density not affected by BC additions | Ferralsol | Woody Eucalyptus 'Blue Mallee' | 550 | [240] |
| Pea (*Pisum sativum*, cv. 'Ashton') | | | | Field Study | | 30 min | |
| Broccoli 'Ironman' | | | | Short-term, 1 year | | 10 t/ha | |

In terms of productivity alone, a majority of the studies reported a beneficial impact of BC on crop yields [53,225,226,229,232–237]. Experimental plants included lettuce, cabbage, radish, tomato, wheat, rice, maize, and okra. Soils were amended with BC derived from major feedstock sources such as hardwood, manure, and crop residues. Positive results from this mixture of plants and biochars indicate a theoretical system to 'mix and match' crop with BC for optimal productivity. Interestingly, none of the studies included perennial plant species in the experimental design. That is another area where the impact of BC remains to be assessed.

Due to the range of tested soil conditions, many factors altered by BC amendment were implicated in reported yield gains. For instance, lettuce yields were positively influenced with 20 and 30 t/ha fecal-derived BC (450 °C) [226]. Mineral-enriched BC proved to be an effective fertilizer for two growing cycles in the greenhouse pot study. Additional experiments in greenhouses with leafy crops proved that significant yield increases are possible with BC soil amendment [20,241]. Two studies with wheat showed increased yields as a result of soils amended with 1–2% apple wood-derived BC (450 °C) due to increased nitrogen levels [232] and increased soil water retention with 8 t/ha date palm tree residue-derived BC (300–600 °C) [229]. The increase in soil organic carbon and the stimulatory effect on microbial communities raised rice yields in soil amended with rice husk BC (350 °C) [235] and okra yields amended with hardwood-derived BC (450 °C) [237]. In addition to reporting increases in yields, these studies also discussed the limitations of field applications of BC.

BC contains key plant nutrients, although at a low level as demonstrated by several studies, and it may have led to the lack of a complete plant nutrient profile in the soils to obtain a desirable increase in yields [225]. Multiple studies reported mixed results in terms of crop production [227,239] or described no effect [166,230,231,240]. Soil nutrient content and CEC were improved with BC amendment but were short-lived and resulted in comparable crop productivity compared to controls in studies with rice and wheat growing in wheat straw BC (350–550 °C) [166], potatoes with green plantain peel-derived BC (450–500 °C) [230], and tomatoes and maize with walnut shell-derived BC (900 °C) [231]. The growth of Spring barley and sunflowers was tested with hardwood-derived BC (550 °C) at 72 t/ha. The treatments increased barley yields but had no effect on sunflower productivity [239]. The BC-only amendment did increase soil water status and carbon levels; however, increased barley productivity was noted only when BC was mixed with NPK compared to NPK-only controls. While increased water-holding capacity and soil carbon levels with peanut shell-derived BC (350 °C) were also reported [227], these alterations did not lead to any effect on lettuce yields.

Undesirable effects on crop productivity following BC soil amendment were also reported in two of the studies [228,238]. Although beans demonstrated an increased rate of germination in BC amended soils, their yields were significantly reduced with hardwood BC (350 °C) application at 2% and 5% [228]. Other studies with legumes reported a gain in yield when grown in BC-amended soils. The yields of mash bean improved with sugarcane bagasse BC (350 °C), with and without chemical fertilizer, due to the increased SOC, total N, and decreased bulk density. Importantly, nitrogen fixation increased by 83% in the biochar-only treatment due to higher nodule numbers [242]. Additionally, fava bean growth with wheat straw-derived BC (500 °C) amendment applied at a 2.5% w/w rate in addition to saltwater irrigation led to significantly increased dry seed yield compared to controls, which was mainly attributed to the high salt sorption capacity of BC [243]. The higher nutrient content in the crop residue-derived BCs reported above may have helped elevate yields compared to controls, while the already nutrient poor hardwood-derived BC may have reduced bean yields. While BC can be a source of nutrients, the complex interactions in the soil environment may have reduced the capacity of available nutrients in the soils inflicting significant yield losses [244]. Additional studies are required to develop a more comprehensive model of BC effect on legume production.

Other factors potentially responsible for lower productivity include soil nutrient deficiencies found with sugarcane bagasse and sunflower-derived BC-amended (500–600 °C) soils [238]. As a result of decreased nitrogen uptake with increasing BC application, sunflower seed and oil yield saw a significant decrease. The 1% field application of the BCs may have increased specific communities of bacteria and

enhanced certain enzyme activity such as urease, which is an important enzyme in soil nitrogen status, as reported by a field study with the addition of sugarcane bagasse biochar (SCBC) [162]. However, fungal communities suffered due to SCBC addition, and final yields of *Brassica chinesis* L. (pak choi) were reduced compared to controls. It was found that a 4% application rate of SCBC supported normal plant growth and increased sugar and cane yields [245]. The SOC, soil–water related properties, and nutrient levels were enhanced by SCBC, leading to increased plant productivity. Further research is needed to identify BCs appropriate for specific plant species and initial soil characteristics for improved plant growth and development.

Although crop responses were generally positive, the high variability within the listed studies makes it difficult to draw any broad conclusions except that the type and application rate of BC will require customization. The benefits of BC application mainly consist of increased water-holding potential, better nutrient cycling, and increased soil carbon reserves. This may lead to no effect or only minimally increasing yields in the short term, but further testing in the field should illuminate the effects of long-term BC amendment on crop yields [246]. Other regions of industrial agriculture and tropical environments may show a more pronounced BC effect and may be better at exploiting the advantages of BC. BC application in marginal soils will likely lead to increased crop productivity by increasing the overall soil fertility through pH and CEC adjustments, better water retention, and increased microbial activity [247,248]. Nevertheless, considerable caution should be observed when using extremely heavy rates of BC. The elevated risk of heavy metal contamination due to feedstocks rich in accumulated metals or other phytotoxic compounds could decrease crop productivity with increasing BC applications [249,250].

The overall conclusion is that BC application is favorable for improving crop productivity sustainably. Certain agricultural systems require different inputs to achieve higher crop yields, and designing BC to meet those specific needs could lead to optimized production methods and products.

## 5. Conclusions and Future Opportunities

Analysis of the published literature supports the role of BC as one of the many viable solutions to soil-related challenges of food production in the face of persistent global issues. While it is not a panacea, the humble porous carbon-composed BC has the ability to physically, biologically, and chemically alter soil properties, which has multifarious consequences. There is an opportunity for carbon sequestration and establishing carbon negative cycles with the expanded use of BC. Countering deteriorating soil health due to industrial agriculture, BC amendment can help support higher crop productivity and contribute to improving global food security. The overall impact of BC can be increased by its application on highly weathered and marginal soils that are characterized by depleted nutrients levels, reduced water retention, and the lack of a competent soil structure.

Future climate models indicate that water stress will be a key driver of reduced crop productivity. BC has been shown to improve soil water-holding capacity, making it a potential candidate for alleviating water stress [251–253]. Finding the right permutation of BC feedstock, application rate, and crop variety are vital for improving agricultural production and reducing the related carbon footprint. Initial progress in understanding the effects of BC use has led to the efforts of producing 'designer biochars' which promise to exploit the positive properties and dampen negative effects mainly through feedstock selection but also through chemical, physical, or natural alterations of the BC [91,254–256].

Various studies and published meta-analyses on BC have pointed out the benefits of BC soil amendment; however, there are still several areas that require attention and resolution. Important aspects that affect crop productivity include feedstock sources, various BC production methods, initial soil characteristics, crop variety, and experimental conditions. Others have also reported irregular effects of BC on crop productivity due to these variables [257]. Since it is an irreversible

decision to amend soils with BC, the various impacts of augmenting soils are important considerations before conducting field applications [258].

Several articles were not included in the comprehensive literature review due to the omission of key variables that are critical for assessing the overall impact of BC studies. The conclusions of this review are that standardizing BC experiments is vitally important to hypothesis testing and replicating studies to move the research field forward. All future BC articles should include meticulous descriptions of biochar feedstock sources, pyrolysis temperature, and retention time. Not covered in this review, but still significant, are the various economic factors to be considered before undertaking any large-scale BC applications, including the production or acquisition of BC, shipping and transporting, and the time and labor required for field application [237,259]. Additionally, the lack of consistent responses of microbial communities to BC amendment highlights a knowledge gap regarding the mechanism by which microbes interact with BC. BC has the potential to decrease the bioavailability and efficiency of some herbicides, which is yet another variable that needs consideration [260]. Further, analysis of the published literature leads to the conclusion that the following areas need further investigation:

- Biomes underrepresented in the current biochar-associated literature, such as forests and perennial crops (the vast majority of BC studies are directed toward temperate and tropical areas);
- Effects of biochar on non-model crop species (present studies primarily focus on model organisms such as tomato, maize, rice, and wheat);
- Evaluation of BC in field studies to build upon the extensive greenhouse studies;
- Develop an understanding of the highly complex interactions between different soil types, different biochar types, and their impact on plant productivity
- Assessment of biochar-amended soil microbial activity through meta-genomics approaches;
- Longer-term experiments to understand characteristics of 'aged' BC to assess its temporally evolving properties in soils;
- Development of cost-effective ways to minimize environmental impacts by incorporating organic fertilizer amendments such as BC.

The multidimensional and complex interactions between inherent soil properties, variable biochar properties depending on the type of feedstock used, the genetic background of the plant, and the limited amount of available empirical data make it almost impossible to predict the outcome of BC amendment. This is an obvious conclusion of this and several previous studies. Knowledge generated from the above-mentioned areas of investigation is expected to enable the large-scale utilization of biochars in agriculture, representing a step toward establishing carbon negative ecosystems.

**Author Contributions:** Conceptualization—A.D., E.T. and R.G.; Resources—A.D.; Writing—review and editing—E.T., R.G., A.D.; Supervision—A.D.; Funding acquisition—A.D. All authors have read and agreed to the published version of the manuscript.

**Funding:** This work was supported in part by the USDA-NIFA Hatch Grant WNP00011 to A.D.

**Acknowledgments:** The authors thank Seanna Hewitt, Karen Adams, and Richard Sharpe for their critical reading of the manuscript and their feedback.

**Conflicts of Interest:** A.D. serves as a consultant for AgEnergy Solutions—A biochar production startup company based in Spokane, WA, USA. AgEnergy Solutions had no role in the design of the study; in the collection, analyses, or interpretation of data; in the writing of the manuscript, or in the decision to publish the results.

## Abbreviations

| | |
|---|---|
| BC | biochar |
| SOM | soil organic matter |
| GHG | greenhouse gas |
| CEC | cation exchange capacity |
| FTIR | Fourier transform infrared spectroscopy |
| SBBC | sugarcane bagasse biochar |
| C | carbon |
| N | nitrogen |
| P | phosphorous |
| K | potassium |
| Ca | calcium |
| Mg | magnesium |
| Fe | iron |
| Na | sodium |
| Si | silicon |
| Cd | cadmium |
| Cu | copper |
| Ni | nickel |
| Pb | lead |
| Zn | zinc |

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
