# Peer review of "Biochar—A Panacea for Agriculture or Just Carbon?"

_horticulturae, doi:10.3390/horticulturae6030037_

Round 1

Reviewer 1 Report

This manuscript is a review of the current state of research on biochar and how biochar affects soil properties from a soil health/agronomic perspective. Previous studies have shown mixed results for biochar application in agronomic settings, so a literature review to systematically frame the current findings and better inform future studies is needed.

The current manuscript eludes to what previous studies, reviews, and meta data analyses have found that biochar results depend on the type of biochar and soil type. The authors have done a good job consulting and including the most recent (up to 2019) studies, but I think this manuscript could benefit from a better framing and synthesis of current studies. What new conclusions can we get from this review? How does either biochar type or the inherent soil properties effect the results?

I've given some specific comments below, especially related to the tables, where I think improvements can be made. Think of what is really needed in the tables, what could be left out, and what might need to be added. Additionally one major item which is often cited for a lack of more widespread adoption of biochar application in agronomic settings is the amount that needs to be applied to see any yield changes or soil improvements. Several of the tables do not include amount of biochar applied, which I think is very important to include. 

Specific comments and suggestions:

Abstract:

General comments:

From the abstract it is not clear whether this is a research paper, review, or meta-data analysis. It should be very clear in the abstract and the reader should not have to read into the body of the paper for this.

L16: not clear what soil performance is.

L21-24: This sentence should be expanded upon, as there is really nothing new provided in the abstract regarding biochar and soils. The general statement: “

 Soil amendment with BC has been shown to have an overall positive impact on soil health and crop productivity; however, initial soil properties need to be considered prior to the application of BC” can be said about any organic amendment. What specific positive impacts? What initial soil properties are important? What is the role of feedstock type say on soil impacts?

Introduction:

L53: Soils do currently store more than twice the amount of atmospheric CO2!

L57-58: Erosion was/is a huge part of declining soil health also. What we see at the surface of many soils currently being farmed is actually the original A horizon mixed with the B horizon, or the totality of the A horizon has been lost and the B horizon is now at the surface.

L63-65: I think there should be a reference or two here to support the claims in the sentence. Manure, compost, and even biosolids have been land applied to many areas without increases in heavy metals. Particularly in the case of class A biosolids, this represents a high-quality waste product which can be an important source of both SOM and plant available nutrients.

L68-69: I would reformulate this as these black earth soils are hypothesized as being formed through long-term application of BC and probably also manure (as indicated by high P values).

L76-78: As already mentioned, the fact that this is a review paper should be evident in the abstract. Also it is missing in the introduction a bit more about uncertainties of biochar that would support a review of the current literature.

L108-109: The threshold bulk density for impacts on root growth will depend on texture, with clayey soils having a lower bulk density threshold.

L119-120: Study 32 is not in the table, and study 33 is listed in the table as having a “slight decrease.” This makes it a bit misleading in the table, as I would assume that slight decrease means there was a statistically significant decrease in bulk density, but the magnitude was not that large.

Table 1: not clear what “selected studies” refers to. In the text, it mentions all studies from 2019. A suggestion, for soil type to instead include maybe %clay or % clay and %sand, as a general textural class can be quite broad. Also for the effects on bulk density, available water content, and porosity, it isn’t clear how these were measured (are they consistent across studies), when an increase/decrease were detected, was this significant? Also compared to an unamended control? What was the time since application and also how much was applied? Is it important to include lab studies and how can bulk density be measured on pots or small cores in the lab? I would probably leave out the lab studies. The country of the study is not really important and could be dropped.

L595-600: What search engine was used for this search? Not clear what the “first 333 entries” refers to? Does this mean that the other ones were discarded? Not clear what “333 entries were reviewed and 155 examined for the further criteria” means”

Table 6:

Consider formatting this table in landscape as opposed to portrait. That will allow putting each entry in one row as opposed to having some columns with multiple subrows (e.g. the country, texture, duration). Consider if country is really necessary, as it doesn’t really add information on the soil or even climatic conditions. As in Table 1, use %clay as opposed to a vague texture class. Also some of the soils listed have the Soil Taxonomy or FAO-WRB soil type and others do not. It should be consistent.

Order the entries either by alphabetical order or maybe separate row crops (e.g. rice, corn, wheat) from vegetable crops. Additionally it might be good to separate field from greenhouse studies.

Author Response

The authors are thankful to the reviewers for their critical and supportive comments. We have considered each comment and have made revisions to the manuscript along with responding to the comments. The suggested revisions have certainly improved the manuscript and we hope that the review will now be acceptable for publication.

Reviewer 1

  1. 1. This manuscript is a review of the current state of research on biochar and how biochar affects soil properties from a soil health/agronomic perspective. Previous studies have shown mixed results for biochar application in agronomic settings, so a literature review to systematically frame the current findings and better inform future studies is needed. The current manuscript eludes to what previous studies, reviews, and meta data analyses have found that biochar results depend on the type of biochar and soil type. The authors have done a good job consulting and including the most recent (up to 2019) studies, but I think this manuscript could benefit from a better framing and synthesis of current studies.

Author Response:

The review article represents the current state of understanding of the role of biochar, which as the reviewer rightly points out, is comprised of mixed results. Building upon the existing studies, the focus of the conclusions therefore has been to identify areas that need further investigation to understand the reason behind the variable responses observed in the cited studies. The lack of standardization in the production of biochar, applications to different soil types, and the genetic background of plants are the key variables that need to be considered.

  1. What new conclusions can we get from this review?

Author Response: The new conclusions have been outlined in section 4 titled Conclusions and Future Directions. Specifically, please see page 44, lined 718 and 728, and the text that follows the latter. Briefly, the conclusions allude to need for standardization of experimental conditions and design, need for field studies and evaluation of biochar in several additional ecological, agricultural, economic and environmental context.

  1. How does either biochar type or the inherent soil properties effect the results?

Author Response: The multidimensional and complex interactions between inherent soil properties, variable biochar properties depending on the type of feedstock used, and the genetic background of the plant, and the limited amount of available empirical data make it almost impossible to predict the outcome of BC amendment. This is an obvious conclusion of this and several previous studies. This sentence has been added to the final paragraph of the conclusions section. Please see page 45, lines 745-748.

  1. I've given some specific comments below, especially related to the tables, where I think improvements can be made. Think of what is really needed in the tables, what could be left out, and what might need to be added. Additionally, one major item which is often cited for a lack of more widespread adoption of biochar application in agronomic settings is the amount that needs to be applied to see any yield changes or soil improvements. Several of the tables do not include amount of biochar applied, which I think is very important to include. 

Author Response: Thank you for the excellent advice regarding the amount of biochar applied in various studies. Indeed, it is a very important point, which we missed to include. The table has been modified as advised.

Specific comments and suggestions:

General comments:

  1. From the abstract it is not clear whether this is a research paper, review, or meta-data analysis. It should be very clear in the abstract and the reader should not have to read into the body of the paper for this.

Author Response: The abstract has been modified as advised to reflect that it is a literature review and includes a focused meta-analysis as well.

  1. L16: not clear what soil performance is.

Author Response: The edited abstract does not mention soil performance.

  1. L21-24: This sentence should be expanded upon, as there is really nothing new provided in the abstract regarding biochar and soils. The general statement: “

Author Response: The abstract has been edited for content as advised.

  1. Soil amendment with BC has been shown to have an overall positive impact on soil health and crop productivity; however, initial soil properties need to be considered prior to the application of BC” can be said about any organic amendment. What specific positive impacts? What initial soil properties are important? What is the role of feedstock type say on soil impacts?

Author Response: The abstract is limited to 200 words, therefore, general rather than specific concepts have been included. However, as advised, productivity has been characterized as yield, and the soil properties have been qualified by indicating the physical, chemical, and biological nature. 

Introduction:

  1. L53: Soils do currently store more than twice the amount of atmospheric CO2!

Author Response: Removed “can” as advised. 

  1. L57-58: Erosion was/is a huge part of declining soil health also. What we see at the surface of many soils currently being farmed is actually the original A horizon mixed with the B horizon, or the totality of the A horizon has been lost and the B horizon is now at the surface.

Author Response: The comment about erosion has been included as advised. Please see page 2, line 68

  1. L63-65: I think there should be a reference or two here to support the claims in the sentence. Manure, compost, and even biosolids have been land applied to many areas without increases in heavy metals. Particularly in the case of class A biosolids, this represents a high-quality waste product which can be an important source of both SOM and plant available nutrients.

Author Response: As advised, two references have been added to address the statements. Please see page 2, line 72.

  1. L68-69: I would reformulate this as these black earth soils are hypothesized as being formed through long-term application of BC and probably also manure (as indicated by high P values).

Author Response: As advised, the sentence has been edited to indicate the role of manure. Please see page 2, line 74.

  1. L76-78: As already mentioned, the fact that this is a review paper should be evident in the abstract. Also, it is missing in the introduction a bit more about uncertainties of biochar that would support a review of the current literature.

Author Response: The edited abstract clarifies that this is a review and includes a focused meta-analysis. As advised, the introduction has been edited to include a rationale for this review. Please see page 3 lines 83-93.

  1. L108-109: The threshold bulk density for impacts on root growth will depend on texture, with clayey soils having a lower bulk density threshold.

Author Response: As suggested, this concept has been included. See page 3 and 4, lines 124 – 128.

  1. L119-120: Study 32 is not in the table, and study 33 is listed in the table as having a “slight decrease.” This makes it a bit misleading in the table, as I would assume that slight decrease means there was a statistically significant decrease in bulk density, but the magnitude was not that large.

Author Response: As advised, the table has been is edited to include all references. Regarding the term slight decrease, the reviewer surmises correct. The table legend has been edited to explain this context.

  1. Table 1: not clear what “selected studies” refers to. In the text, it mentions all studies from 2019. A suggestion, for soil type to instead include maybe %clay or % clay and %sand, as a general textural class can be quite broad. Also, for the effects on bulk density, available water content, and porosity, it isn’t clear how these were measured (are they consistent across studies), when an increase/decrease were detected, was this significant? Also compared to an unamended control? What was the time since application and also how much was applied? Is it important to include lab studies and how can bulk density be measured on pots or small cores in the lab? I would probably leave out the lab studies. The country of the study is not really important and could be dropped.

Author Response: The suggestion provided by the reviewer is excellent, however the type of soil reported is as per the original citation and the percentage breakdown has not been provided. Further, the reviewer correctly points out the lack of consistency in reporting the data. All studies included a control. As advised, the column related to country has been removed. A column reporting rates of interest has been included. Currently, lab studies predominate in the literature, and therefore important for inclusion.

  1. L595-600: What search engine was used for this search? Not clear what the “first 333 entries” refers to? Does this mean that the other ones were discarded? Not clear what “333 entries were reviewed and 155 examined for the further criteria” means”

Author Response: The Google Scholar search engine was used as indicated on page 22, line 620. All the criteria are listed on lines 618 – 622.  The search criteria has been clarified. Please see page 22, lines 618 – 622.

  1. Table 6:

Consider formatting this table in landscape as opposed to portrait. That will allow putting each entry in one row as opposed to having some columns with multiple subrows (e.g. the country, texture, duration). Consider if country is really necessary, as it doesn’t really add information on the soil or even climatic conditions. As in Table 1, use %clay as opposed to a vague texture class. Also, some of the soils listed have the Soil Taxonomy or FAO-WRB soil type and others do not. It should be consistent. Order the entries either by alphabetical order or maybe separate row crops (e.g. rice, corn, wheat) from vegetable crops. Additionally, it might be good to separate field from greenhouse studies.

Author Response: As advised, the orientation of the table has been changed to landscape, and other relevant edits incorporated. As in case of other studies, the soil type listed is as per the original publication. In fact, this is another area that needs standardization.

Reviewer 2 Report

The manuscript provides an overview of biochar impacts on soil properties by summarizing independent research on the topic. Overall the paper is thorough in its treatment of the subject. 

In terms of content I have little feedback. My only suggestion is that I think all reviews should include a short method section as to how they identified articles for inclusion, what articles are excluded (for instance most all reviews only consider papers written in their native language), etc. While the PRISMA criteria on reporting is designed for meta-analyses and systematic reviews, I think it is good practice to include that for all reviews so the readers can have a clear understanding of article inclusion and exclusion. 

Also, given the large amount of reviews on biochar that have come out, I think it would be very worthwhile taking a sentence or two indicating how this one is different from the (at least) two reviews in 2019 that covered the same topic. 

In terms of style I felt that the paper almost read like an annotated bibliography, with each independent study summarized, rather than overall trends summarized and then supported by individual studies. There is nothing wrong with the way it is presented, but as a reader it became very tedious to read this paper. Along the same lines, I did not find the tables very effective or informative.  Again, I know that this is a review and not a meta-analysis, but I feel that more could be done to combine studies into categorical responses to help the reader instead of saying "this study raised nutrients, this one lowered it, this one raised it, this one raised it." 

The paper is well written in terms of language and grammar. Again, it is a comprehensive treatment of the topic. My suggestions are mainly in regards to the style and readability.

Author Response

The authors are thankful to the reviewers for their critical and supportive comments. We have considered each comment and have made revisions to the manuscript along with responding to the comments. The suggested revisions have certainly improved the manuscript and we hope that the review will now be acceptable for publication.

Reviewer 2

  1. The manuscript provides an overview of biochar impacts on soil properties by summarizing independent research on the topic. Overall the paper is thorough in its treatment of the subject. 

Author Response: We thank the reviewer for a positive summary of this manuscript.

  1. In terms of content I have little feedback. My only suggestion is that I think all reviews should include a short method section as to how they identified articles for inclusion, what articles are excluded (for instance most all reviews only consider papers written in their native language), etc. While the PRISMA criteria on reporting is designed for meta-analyses and systematic reviews, I think it is good practice to include that for all reviews so the readers can have a clear understanding of article inclusion and exclusion. 

Author Response: As advised the methodology has been clarified. Please see page 22, lines 618 – 622.

  1. Also, given the large amount of reviews on biochar that have come out, I think it would be very worthwhile taking a sentence or two indicating how this one is different from the (at least) two reviews in 2019 that covered the same topic. 

Author Response: As advised, the last paragraph of the introduction section (please see page 3, lines 83-93) highlights how this review is different.

4. In terms of style I felt that the paper almost read like an annotated bibliography, with each independent study summarized, rather than overall trends summarized and then supported by individual studies. There is nothing wrong with the way it is presented, but as a reader it became very tedious to read this paper. Along the same lines, I did not find the tables very effective or informative.  Again, I know that this is a review and not a meta-analysis, but I feel that more could be done to combine studies into categorical responses to help the reader instead of saying "this study raised nutrients, this one lowered it, this one raised it, this one raised it." 

Author Response: As advised, the tables have been modified to properly summarize the studies and also increase their utility. The reviewer rightly points out that comprehensive coverage of so many reports can become somewhat repetitive.

5. The paper is well written in terms of language and grammar. Again, it is a comprehensive treatment of the topic. My suggestions are mainly in regards to the style and readability.

Author Response: We appreciate the supportive comments.